# The Origin and Immune Recognition of Tumor-Specific Antigens

**DOI:** 10.3390/cancers12092607

**Published:** 2020-09-12

**Authors:** Anca Apavaloaei, Marie-Pierre Hardy, Pierre Thibault, Claude Perreault

**Affiliations:** Institute for Research in Immunology and Cancer, Université de Montréal, Montréal, QC H3T 1J4, Canada; anca.apavaloaei@umontreal.ca (A.A.); marie-pierre.hardy@umontreal.ca (M.-P.H.)

**Keywords:** antigen processing and presentation, cancer immunotherapy, cross-priming, immunogenicity, major histocompatibility complex, T lymphocyte, tumor-infiltrating lymphocytes, tumor microenvironment, tumor-specific antigen

## Abstract

**Simple Summary:**

Cancer immunology is a rapidly evolving field. In this context, this review article has three objectives. First, to explain the genomic origin of tumor antigens and to emphasize that many of them are encoded by unconventional RNAs. Second, to discuss the inherent limitations of all strategies aimed at discovering tumor antigens, and to highlight the importance of using mass spectrometry validation for each antigen considered for clinical trials. Third, to explain that many tumor antigens are not spontaneously detected by the immune system, because they are not presented adequately by dendritic cells. Concepts presented in this article must be taken into account in the design of cancer immunotherapies and of cancer vaccines in particular.

**Abstract:**

The dominant paradigm holds that spontaneous and therapeutically induced anti-tumor responses are mediated mainly by CD8 T cells and directed against tumor-specific antigens (TSAs). The presence of specific TSAs on cancer cells can only be proven by mass spectrometry analyses. Bioinformatic predictions and reverse immunology studies cannot provide this type of conclusive evidence. Most TSAs are coded by unmutated non-canonical transcripts that arise from cancer-specific epigenetic and splicing aberrations. When searching for TSAs, it is therefore important to perform mass spectrometry analyses that interrogate not only the canonical reading frame of annotated exome but all reading frames of the entire translatome. The majority of aberrantly expressed TSAs (aeTSAs) derive from unstable short-lived proteins that are good substrates for direct major histocompatibility complex (MHC) I presentation but poor substrates for cross-presentation. This is an important caveat, because cancer cells are poor antigen-presenting cells, and the immune system, therefore, depends on cross-presentation by dendritic cells (DCs) to detect the presence of TSAs. We, therefore, postulate that, in the untreated host, most aeTSAs are undetected by the immune system. We present evidence suggesting that vaccines inducing direct aeTSA presentation by DCs may represent an attractive strategy for cancer treatment.

## 1. Introduction

The introduction of immune checkpoint therapy in the treatment of several cancer types has dramatically changed the landscape of oncology [1,2]. The success of this approach is based on the paradigm that T lymphocytes, and in particular the CD8 subset [3], recognize tumor antigens that can elicit vigorous immune responses and tumor rejection [4]. Attention has focused on major histocompatibility complex (MHC)-associated peptides (MAPs), which are the ligands recognized by classic T cells. However, the precise nature of the MAPs capable of causing tumor rejection remains unclear. Seen as promising for decades, tumor-associated antigens (TAAs) have recently fallen into disfavor. TAAs are MAPs that are not cancer-specific but are overexpressed in cancer cells (Table 1). Since they are part of the normal immune self, TAAs are not highly immunogenic and TAA vaccines have yielded disappointing results [5,6]. Strong evidence suggests that anti-tumor immune responses potentiated by immune checkpoint therapy are directed against tumor-specific antigens (TSAs); that is, MAPs found only on cancer cells [4,7,8]. Nonetheless the molecular landscape of actionable TSAs remains largely elusive. 

## 2. Misconceptions about TSAs

### 2.1. Neoantigens and the Fallacy of the Converse

Efforts seeking to discover TSAs initially focused on MAPs coded by mutated exons. This makes sense, because the cancer specificity of mutated MAPs is unquestionable, and exons have long been considered the sole protein-coding genomic sequences. These efforts led to the discovery of mutated TSAs (mTSAs), only a few of which were validated by mass spectrometry [4,20]. Furthermore, in several cases, tumor-infiltrating lymphocytes specific for mTSAs were shown to have the ability to mediate tumor regression [21,22]. Unfortunately, excitement over the discovery of exonic mTSAs led to a misconception with major implications. As exonic mTSAs were, in selected cases, sufficient to elicit anti-tumor responses, it was assumed that they were necessary to elicit anti-tumor responses (fallacy of the converse). In other words, it was postulated that exonic mTSAs were the sole actionable TSAs. This reasoning was incorrect, because the term TSA should designate any cancer-specific MAP, whatever its genomic origin (exonic or not) and irrespective of its mutational status (i.e., mutated or not). This concept has important implications. First, annotated exons represent only 2% of the genome and many allegedly non-protein coding (non exonic) sequences are coding for proteins and do generate MAPs [23,24,25,26]. Second, epigenetic and splicing aberrations as well as frameshift translation in cancer cells lead to the appearance of numerous proteins and MAPs that are not found in normal cells. Cancer-specific MAPs resulting from translation of any open reading frames not expressed in normal adult cells are referred to as aberrantly expressed TSAs (aeTSAs) (Table 1). aeTSAs can derive from i) canonical (annotated) onco-fetal genes that are normally repressed in the adult organism (e.g., MAGEA3) or ii) from non-canonical transcripts that arise from cancer-specific epigenetic changes, frameshift translation, or splicing aberrations. Notably, translation of a whole cancer-specific transcript can yield more numerous TSAs than a single base pair substitution [27]. When compared to mTSAs, aeTSAs display two advantageous features. First, they are more numerous [18]. Indeed, in a recent study of 23 ovarian cancers, 103 TSAs were identified of which only three were exonic mTSAs [19]. Second, whereas mTSAs are generally unique to individual patients, aeTSAs are shared by many tumors. In ovarian cancer, 78% of transcripts coding for individual aeTSAs were found in at least 10% of tumors and 18% in at least 80% of tumors [19]. 

When exonic mTSAs were discovered, they were frequently labeled as neoantigens. In fact, the terms TSA and neoantigens should be synonymous. Accordingly, we would have no objection to talk of neoantigens and to classify them into mutated and aberrantly expressed neoantigens. However, many scientists still believe that neoantigens means exonic mTSAs. Therefore, as recommended by Haen et al. [5], we will refrain from using the term neoantigen in order to avoid any ambiguity.

### 2.2. Can mTSAs Be Identified without Mass Spectrometry Analyses?

Mass spectrometry remains the only method that allows direct and definitive identification of the amino acid sequence of MAPs and TSAs [5,28,29,30]. However, mass spectrometry analyses require (i) large tumor samples and (ii) specialized equipment and expertise, which are not widely available. Hence, research teams have tried to identify TSAs using genomic data (exome and/or transcriptome sequencing) and algorithms to predict MHC-binding affinity. Unfortunately, two types of evidence suggest that most “predicted TSAs” are false discoveries: one type is based on mass spectrometry validation of predicted TSAs, the second on in-depth genomic analyses. Almost all studies have focused exclusively on exonic mTSAs. 

#### 2.2.1. Mass Spectrometry Validation of Predicted mTSAs

In 16 primary human hepatocellular carcinomas, Löffler et al. predicted the occurrence of 1888 exonic mTSAs (a mean of 118 per tumor), none of which was validated by mass spectrometry [13]. In colorectal carcinomas, Newey et al. predicted the occurrence of 304 mTSAs, of which only three were validated by mass spectrometry. In smaller scale studies, not a single mTSA was validated by mass spectrometry analyses of four acute lymphoblastic leukemias [18] and three pancreatic adenocarcinomas [16]. 

#### 2.2.2. In-Depth Genomic Analyses

According to the tenets of immunoediting [31], exonic mTSAs should be under negative selection pressure (enforced by TSA-responsive T cells). This negative selection should decrease the ratio of non-synonymous over synonymous mutations in mTSA-coding sequences. However, no such decrease was found in analyses of genomic data from 8,683 tumor samples [32]. Likewise, comprehensive analyses of over 1,000 melanoma exomes revealed no evidence of HLA-restricted negative selection against exonic mutations [33]. Furthermore, response to immune checkpoint therapy in patients with lung cancer did not correlate more with predicted mTSAs than with the global mutation load [34]. These data mean that the number of exonic mTSAs has been grossly overestimated in many studies and that the mTSA repertoire of a tumor cannot be predicted with current algorithms. The reason for this is that while these algorithms can accurately predict with reasonable accuracy the MHC-binding affinity of a peptide, they fail to take into account the numerous translational and post-translational events that regulate MAP biogenesis and presentation [35,36,37].

#### 2.2.3. Can Reverse Immunology Eliminate False Positive TSA Predictions?

Testing the immunogenicity of validated TSAs using various in vitro methods (MHC-MAP multimers binding, cytokine production, etc.) provides useful information. It shows which TSAs are more likely to stimulate anti-tumor responses in vivo. However, it is commonly assumed that if a predicted mTSA (not validated by mass spectrometry) can elicit T-cell responses from peripheral blood mononuclear cells, it is more likely to be a genuine TSA. We disagree with this assumption. The fact that a predicted TSA is immunogenic simply means that it can be recognized by some T cells; this does not increase the likelihood that this predicted TSA is a genuine TSA (present on cancer cells). Two “peptide stories” illustrate this point: those of E*L*AGIGILTV and RIAECILGM. The E*L*AGIGILTV peptide is an in vitro modified version of the wild-type E*A*AGIGILTV MART-1/Melan-A26-35 decamer. Hence, for the immune system, E*L*AGIGILTV is akin to an mTSA. While this peptide is not found on cancer cells, it is so immunogenic that it is commonly used as a positive control in ex vivo immunogenicity assays [38,39]. The TEL-AML1 fusion protein results from a 12:21 chromosomal translocation and is frequently found in B-cell precursor acute lymphoblastic leukemia. A peptide resulting from this fusion protein, RIAECILGM, was predicted to be presented by HLA-A^*^02:01, and priming of T cells against this peptide generated cytotoxic T cells that killed autologous leukemic cells [40]. Further in-depth studies showed that this epitope is not presented by leukemic cells; it is not endogenously processed, because it is cleaved by proteasomes [41]. Killing of leukemic cells by T cells primed against RIAECILGM was most likely due to the inherent cross-reactivity of T cells, which is further amplified in T-cell lines [42]. Indeed, positive selection in the thymus preferentially rescues cross-reactive T cells [43], and a single T-cell receptor may recognize more than a million different MAPs [44]. 

## 3. Strategies for Mass Spectrometry-Based Identification of aeTSAs

aeTSAs present several attractive features. They are more common than mTSAs, and they are shared by many tumors of a given type (Table 1). Furthermore, in pre-clinical models, they were shown to elicit curative anti-tumor responses [18,45]. Since aeTSAs can be coded by any reading frame of the entire genome, their search space is greater than that of mTSAs [8,46,47]. Therefore, it is currently impossible to rely on available bioinformatic tools to predict the aeTSA landscape of a tumor, and mass spectrometry analyses are mandatory for aeTSA identification. The key question here is: once a putative aeTSA is identified, how do we demonstrate its cancer-specificity? In other words, how can we prove that an unmutated MAP is not expressed by any normal cell type? Three approaches have been developed, each with pros and cons.

The first approach postulates that a MAP is a TSA if it is found in cancer cells but not in an atlas of MAPs identified in normal tissue extracts [9,10]. The problem here is that this atlas does not contain the MAP repertoire of all cell types. Thus, since most epithelial cells express lower levels of MHC molecules than hematopoietic cells [48,49], whole tissue extracts are enriched in hematopoietic relative to epithelial MAPs. Furthermore, several cell types in the organism are not present in numbers sufficient for mass spectrometry analyses. Hence, some TSAs identified with this approach may be false positives. 

In the second approach, normal adjacent tissue (not tumor-infiltrated) is used as a negative control [50]. Once again, the absence of a MAP in the normal adjacent tissue does not guarantee that it is not present in other cell types in the organism. We speculate that this approach can also lead to dismissal of genuine TSAs. Our assumption is based on the notion that normal tumor-adjacent tissue may in fact not be normal but rather pre-neoplastic and, therefore, share TSAs with the tumor [51]. Indeed, as we age, physiologically healthy tissues such as skin [52,53], colon [54,55], esophagus [56,57], and blood [58,59,60,61,62,63,64,65] acquire mutations in cancer-associated genes. Timing analyses suggest that driver mutations often precede diagnosis by many years, if not decades. A notable example is ovarian adenocarcinoma, which appears to have a median latency of more than 10 years [66]. 

The third approach is based on the assumption that a TSA cannot be present in cells that do not express TSA-coding transcripts. In contrast to mass spectrometry analyses, transcriptomic analyses have been performed in many subjects on practically all cell types. Hence, when we identify aeTSA candidates, we evaluate whether its coding transcript is found in normal tissues from the GTEx database (https://gtexportal.org/home/) or in our datasets of medullary thymic epithelial cells [26]. We believe that inclusion of medullary thymic epithelial cells in the “negative controls” is important for three reasons: (i) they orchestrate central immune tolerance [67], (ii) they express much higher levels of MHC I molecules than other epithelial cell types [49], and (iii) they promiscuously express more genes than other types of somatic cells [68,69]. Promiscuous gene expression in medullary thymic epithelial cells involves not only classic genes, but also other genomic regions such as endogenous retroelements [26]. The downside of this approach is that it can lead to the dismissal of genuine aeTSAs. Indeed, expression of a transcript in some normal cell does not necessarily lead to expression processing and presentation of the corresponding TSA. 

It must nonetheless be acknowledged that mass spectrometry studies come with intrinsic challenges and limitations [5]. First and foremost, in discovery mode, “shotgun mass spectrometry” has limited sensitivity and, therefore, requires large amounts of starting material for in-depth coverage of the immunopeptidome (e.g., 1 g of tumor tissue). Second, relative to transcriptome sequencing, mass spectrometry has a relatively low throughput and is not quantitative. Finally, mass spectrometry fails to differentiate between isobaric amino acids (Leucine vs. Isoleucine) and is relatively costly in terms of reagents and resources. Several technical innovations are being developed in order to overcome these limitations [30,70]. 

## 4. Immune Recognition of TSAs

### 4.1. Cancer Cells Are Poor T-Cell Activators

The general rules of T-cell priming also apply to cancer cells. Indeed, T-cell recognition of tumors requires both signal 1 (TCR ligands such as TSAs) and signal 2 (co-stimulation) [71]. The most critical positive co-stimulatory signal is provided by CD28 upon interaction with its ligands of the B7 family (CD80/86) on antigen-presenting cells (APCs) [72]. Tumor cells are poor APCs: they express no/low levels of CD28 ligands, and carcinomas (90% of cancers) derive from epithelial cells expressing 10 to 100-fold less MHC I molecules than DCs [49]. As a result, tumor cells are inefficient at directly priming naïve CD8 T cells, and activation of T cells against tumor antigens depends on cross-presentation by professional APCs [73]. Accordingly, anti-tumor responses, either spontaneous or induced with immune checkpoint therapy, correlate with intratumoral infiltration and maturation of cross-presenting CD8α^+^CD103^+^ dendritic cells (DCs) [20,74,75]. These specialized DCs internalize and cross-present tumor antigens to T cells and induce a CD28-dependent proliferation of tumor-specific T cells, which regulates the strength of the immune response [76,77,78,79].

### 4.2. Cross-Presentation Yields a Biased Representation of the TSA Repertoire

The rules governing direct presentation and cross-presentation are different. Direct presentation favors short-lived and rapidly degraded proteins, many of which represent unstable defective ribosomal products that may derive from specialized ribosomes (immunoribosomes) [80,81,82]. In contrast, cross-presentation of exogenous antigens preferentially samples long-lived, stable proteins [83,84]. Thus, direct presentation correlates with the rate of protein translation and proteasomal degradation, whereas cross-presentation correlates with steady-state protein amounts [85]. APCs acquire proteins from donor cells (e.g., cancer cells) through endocytic mechanisms of which the most efficient is phagocytosis [86]. Internalized proteins can then be degraded by proteasomes, either in endocytic organelles or in the cytosol [86,87]. A key implication is that cross-presentation can only display a fraction of TSAs; that is, TSAs derived from highly abundant and stable proteins. Hence, the immune system remains ignorant of TSAs found in unstable and rapidly degraded proteins. 

### 4.3. The Strength of Effector T-Cell Responses

The amplitude of anti-tumor T-cell responses depends on two factors: (i) epitope density on APCs and cancer cells, and (ii) the frequency of antigen-responsive T cells in the pre-immune repertoire. Epitope density (number of MAPs per cell) on APCs during initial priming regulates not only the magnitude but also the avidity and functionality of the effector T-cell population [88,89]. For most—though not all—antigens, cross-presentation yields a lower epitope density than direct presentation [90]. In addition, epitope density on tumor cells dictates their susceptibility to CD8 T-cell cytotoxicity. At this point, intratumoral heterogeneity has to be taken into consideration, because it is a hallmark of all cancers. For immuno-oncologists, this means that individual tumor cells may display different levels of TSA expression. Both in mice and humans, the proportion of TSA-positive tumor cells positively regulates the outcome of interactions between CD8 T cells and the tumor [91,92,93]. The presence of some TSA-negative tumor cells does not necessarily lead to immunotherapy failure. Indeed, TSA-negative tumor cells can be eradicated by T-cell targeting of the tumor stroma and, in particular, endothelial cells [94]. Intratumoral T cells can damage the tumor vasculature via two mechanisms: (i) killing of endothelial cells that cross-present TSAs and (ii) via the potent angiostatic effect of IFN-γ and TNF-α [89,95,96]. Nonetheless, these data suggest that chances of success of immunotherapy should be improved by selecting clonal TSAs (present on most/all cancer cells) and targeting multiple TSAs. 

### 4.4. Vaccination-Induced T-Cell Priming

DC-based vaccines can elicit strong anti-tumor responses against TSAs that are ignored when presented solely by cancer cells (Figure 1). This is illustrated well by the aeTSA VNYLHRNV. While this peptide is expressed at high levels on EL4 cells (908 copies per cell), immunization with irradiated EL4 cells does not prolong the survival of mice upon subsequent injection of unirradiated EL4 cells. However, 100% of mice immunized with DCs coated with VNYLHRNV survive when injected with EL4 cells [18]. Moreover, immunization generates TSA-responsive memory CD8 T cells, since mice survive a novel injection of EL4 cells 100 days later. Hence, VNYLHRNV is highly immunogenic when presented by DCs, but in the absence of therapeutic vaccination, this EL4 TSA is not cross-presented by DCs in vivo. 

Properly designed nanoparticulate liposomal RNA vaccines are efficiently taken up by DCs in secondary lymphoid organs in vivo [97]. In these conditions, since the DC-targeted RNAs drive synthesis of antigenic peptides inside DCs, their processing follows the rule of direct presentation as opposed to cross-presentation. A phase I trial evaluating a nanoparticulate liposomal RNA vaccine in immune checkpoint therapy-experienced melanoma patients (stage III B, C and stage IV) recently provided suggestive evidence that, when presented by DCs, aeTSAs can elicit anti-tumor responses [98]. This vaccine contained four antigens that were originally labeled as TAAs. However, while one of these antigens is clearly a TAA (*TYR*), the other three are probably aeTSAs coded by conventional annotated genes: *MAGEA3*, *CTAG1B* (*NY-ESO-1*), and *TPTE*. Indeed, while *TYR* is expressed in the normal skin, the three putative aeTSAs are not expressed in normal extra-thymic tissues (except for the testis). However, *MAGEA3*, *CTAG1B,* and *TPTE* RNAs are expressed at very low levels in medullary thymic epithelial cells (˂0.5 transcripts per kilobase million [26]). The probability that transcripts expressed at such levels might generate MAPs is very low [35,99] but cannot be formally excluded in the absence of immunopeptidomic studies on medullary thymic epithelial cells. We, therefore, consider that these three antigens are probably, but not certainly, aeTSAs. Notably, objective anti-tumor response correlated more closely with the expansion of T cells recognizing MAGEA3 and CTAG1B [98]. This study, therefore, (i) supports the immunogenicity of aeTSAs in humans and (ii) suggests that direct aeTSA presentation by DCs activates and expands a pool of complementary aeTSA-specific T cells that were insensitive to immune checkpoint therapy and likely tumor-naïve. 

### 4.5. Combining Vaccines and Immune Checkpoint Therapy 

In general, a high density of tumor-infiltrating lymphocytes positively correlates response to immune checkpoint therapy [100]. This suggests that immune checkpoint therapy works at least in part by invigorating T cells responding to cross-presented TSAs. Likewise, preliminary evidence suggests that immune checkpoint therapy may potentiate T-cell response to aeTSAs directly presented by DCs [98]. The idea of combining vaccines and immune checkpoint therapy, therefore, appears very attractive. 

## 5. Conclusions

Therapeutic vaccines can induce durable regressions of premalignant oncogenic human papilloma virus type 16-induced anogenital lesions [101]. To the best of our knowledge, this vaccine containing viral peptides remains the sole therapeutic TSA-based vaccine that has shown reliable efficacy. The first non-viral TSA vaccines tested in clinical trials were based on predicted mTSAs not validated by mass spectrometry [102,103,104,105]. Evidence that most predicted mTSAs not validated by mass spectrometry may be false discoveries does not bode well for the success of these studies. aeTSAs present attractive features and give encouraging results in pre-clinical models. However, evidence supporting their value or superiority over TAAs in humans remains anecdotal and has yet to be formally assessed. For cancer immunologists wishing to develop therapeutic vaccines, the time is not for celebrations but rather to develop innovative research strategies in a climate tinged with both optimism and critical thinking. We also propose that TSAs should be validated by mass spectrometry analyses of primary human tumors before they are tested in clinical trials. Finally, we strongly encourage the sharing of mass spectrometry datasets via the SysteMHC Atlas whose primary objective is to provide a systems-level definition of MAP and TSA repertoires presented by normal and neoplastic cells [30,106]. 

## Figures and Tables

**Figure 1 cancers-12-02607-f001:**
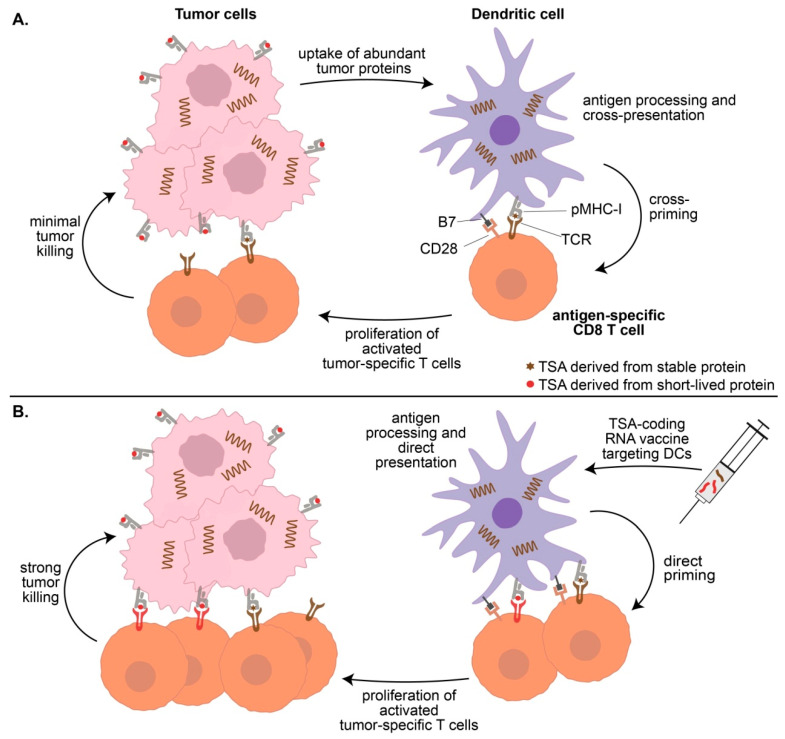
Priming of anti-tumor CD8^+^ T cells by dendritic cells (DCs). (**A**) Most cancer cells are poor antigen-presenting cells (APCs) that are not efficient at direct antigen presentation. DCs are potent APCs, but under basal conditions, they can cross-present only a fraction of the tumor-specific antigens (TSA) repertoire generated by cancer cells. TSAs derived from unstable rapidly degraded proteins (the most common of TSAs) are not cross-presented by DCs and are, therefore, ignored by the immune system. (**B**) Therapeutic mRNA vaccines can deliver any TSA-coding transcripts to DCs for direct presentation to CD8 T cells. In this way, TSAs derived from both short-lived proteins and stable proteins can be detected by CD8 T cells.

**Table 1 cancers-12-02607-t001:** Features of tumor antigens.

Feature	TAAs	mTSAs	aeTSAs
Cancer-specific	No	Yes	Yes
Mutation	No	Yes	No
Shared among tumors	Yes	No	Yes
Number per tumor	Medium-High	Very low	Medium-High
Selected studies containing MS analyses	[9,10,11,12]	[13,14,15,16,17]	[18,19]

aeTSA, aberrantly expressed tumor-specific antigen; MS, mass spectrometry; mTSA, mutated tumor-specific antigen; TAA, tumor-associated antigen.

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
