# Peer review of "The Origin and Immune Recognition of Tumor-Specific Antigens"

_cancers, 2020, doi:10.3390/cancers12092607_

Round 1

Reviewer 1 Report

Apavaloaei et al. present a well written and timely review on the origin and immune recognition of tumor specific antigens.

After a short introduction the authors first point out, in section 2.1, the common misconceptions about tumor specific antigens (TSA), i.e. that most TSAs derive from exonic mutations (mTSAs). Subsequently they stress the importance of non-exonic TSAs that derive from aberrant expression (aeTSAs). These aeTSAs can derive from a) canonical oncofetal genes, normally repressed in the adult organism; or b) from non-canonical transcripts that arise from cancer-specific epigenetic and splicing aberrations. Advantages of aeTSAs compared to mTSAs include their higher frequency and the fact that they may be shared by many tumors.

In section 2.2, the authors argue that mass spectrometry (MS) based analyses are required to demonstrate that predicted TSAs are indeed good substrates for MHC class I presentation. Main reason is that while most algorithms can predict MHC-binding affinity of a peptide, they fail to take into account the many events that regulate the MHC associated expression of peptides.

In section 3, strategies for MS based identification of aeTSAs are discussed. First, the authors state that ‘…, it is currently impossible to rely on available bioinformatic tools to predict the aeTSA landscape of a tumor, and MS analyses are mandatory for aeTSA identification’. Then they move on and state that once a putative aeTSA has been identified and shown to also be a MAP (MHC associated peptide) the main issue is to prove that the unmutated MAP is not expressed by any normal cell type. The authors summarize the pros and cons of three approaches to do this.

In section 4, important aspects of immune recognition of TSAs are discussed, focusing on the fact that cancer cells are poor T-cell activators resulting in dependence of TSA on cross-presentation, yielding a biased representations of TSA repertoire. Then they discuss several factors affecting the strength of the effector T-cell responses, including selection of clonal TSAs and targeting multiple TSAs. Then the potential of vaccination induced T cell priming with or without combination with checkpoint inhibition is discussed.

In their conclusion, the authors stress again the importance of TSA validation by MS based analysis of primary human tumor tissue.

Comments:

Section 2: the authors state (see also Table 1) that mTSA are not shared among tumors. While true for single base pair substitutions that at best can alter 1 amino acid, the more recently described frame shift mutations may potentially yield neoantigens that  are shared between tumors (Koster & Plasterk. Sci Rep 2019;9:6577). Could the authors comment on this.

Section 3: as the authors rightfully stress the importance of MS based validation of TSA, it may be helpful to shortly describe for the reader the main problems encountered in the MS identification of MHC associated peptides in human tumor tissues (sensitivity issues, isobaric aminoacids, etc.).

Author Response

First and foremost, we thank the reviewers for their thoughtful comments and suggestions.

Reviewer 1

Section 2: the authors state (see also Table 1) that mTSA are not shared among tumors. While true for single base pair substitutions that at best can alter 1 amino acid, the more recently described frame shift mutations may potentially yield neoantigens that  are shared between tumors (Koster & Plasterk. Sci Rep 2019;9:6577). Could the authors comment on this.

We agree with the reviewer: frame shift translation represents a potentially rich source of TSAs. This is now explained in lines 75-82, and we now cite the highly relevant study of Koster & Plasterk.

Section 3: as the authors rightfully stress the importance of MS based validation of TSA, it may be helpful to shortly describe for the reader the main problems encountered in the MS identification of MHC associated peptides in human tumor tissues (sensitivity issues, isobaric aminoacids, etc.).

We have now summarized these limitations in lines 185-192: “It must nonetheless be acknowledged that mass spectrometry studies come with intrinsic challenges and limitations [5]. First and foremost, in discovery mode, “shotgun mass spectrometry” has limited sensitivity and therefore requires large amounts of starting material for in-depth coverage of the immunopeptidome (e.g., 1 g of tumor tissue). Second, relative to transcriptome sequencing, mass spectrometry has a relatively low throughput and is not quantitative. Finally, mass spectrometry fails to differentiate between isobaric amino acids (Leucine vs. Isoleucine) and is more costly in terms of reagents and resources. Several technical innovations are being developed in order to overcome these limitations [30,70].”

Reviewer 2 Report

The authors give an interesting insight in types of tumor-specific antigens, in how to detect them and which are interesting for vaccination purposes. Overall, the manuscript is well written.

Comments

1) The high amount of abbreviations that are used make the manuscript difficult to read.

I suggest not to abbreviate 'mutated' and 'aberrantly expressed', neither MAP, mTEC, HPV, TIL, ICT, MS

2) One of the main messages of the paper is that mass spectrometry is of added value to the detection of tumor-specific antigens. This has currently not been written in the abstract text yet. Can the authors please also include a critical discussion on using mass spectrometry for this purpose. What are the hurdles and challenges?

3) 'We present evidence suggesting that vaccines inducing direct aeTSA presentation by DCs represent an attractive strategy for cancer treatment.'

Can the authors please critically discuss if the results are better than when using TAA or mTSA? The text on this subject is quite anecdotical at this moment.

Author Response

First and foremost, we thank the reviewers for their thoughtful comments and suggestions.

Reviewer 2

1) The high amount of abbreviations that are used make the manuscript difficult to read. I suggest not to abbreviate 'mutated' and 'aberrantly expressed', neither MAP, mTEC, HPV, TIL, ICT, MS.

As suggested by the reviewer we have reduced the amount of abbreviations by taking out the following abbreviations: mTEC, HPV, TIL, ICT and MS.

2) One of the main messages of the paper is that mass spectrometry is of added value to the detection of tumor-specific antigens. This has currently not been written in the abstract text yet. Can the authors please also include a critical discussion on using mass spectrometry for this purpose. What are the hurdles and challenges?

The importance of mass spectrometry is now mentioned in the abstract and its main limitations are explained in lines 185-192: “It must nonetheless be acknowledged that mass spectrometry studies come with intrinsic challenges and limitations [5]. First and foremost, in discovery mode, “shotgun mass spectrometry” has limited sensitivity and therefore requires large amounts of starting material for in-depth coverage of the immunopeptidome (e.g., 1 g of tumor tissue). Second, relative to transcriptome sequencing, mass spectrometry has a relatively low throughput and is not quantitative. Finally, mass spectrometry fails to differentiate between isobaric amino acids (Leucine vs. Isoleucine) and is more costly in terms of reagents and resources. Several technical innovations are being developed in order to overcome these limitations [30,70].”

3) 'We present evidence suggesting that vaccines inducing direct aeTSA presentation by DCs represent an attractive strategy for cancer treatment.' Can the authors please critically discuss if the results are better than when using TAA or mTSA? The text on this subject is quite anecdotical at this moment.

In agreement with the reviewer’s comment, we now have included the following statement in the last paragraph of the manuscript (lines 289-291): “However, evidence supporting their value or superiority over TAAs in humans remains anecdotal and has yet to be formally assessed.”